**Data Availability Statement:** All relevant data are within the paper and its Supporting Information files.

**Funding:** This work was supported by the General Research Fund [HKBU12103817] to CKCW,

# Effects of stanniocalcin-1 overexpressing hepatocellular carcinoma cells on macrophage migration

**Cherry C. T. Leung, Chris K. C. Wong** *

Croucher Institute for Environmental Sciences, Department of Biology, Hong Kong Baptist University, Hong Kong SAR, China

* ckcwong@hkbu.edu.hk

## Abstract

Human stanniocalcin-1 (STC1) is a glycoprotein known to participate in inflammation and tumor progression. However, its role in cancer-macrophage interaction at the tumor environment is not known. In this study, the co-culture of the human metastatic hepatocellular carcinoma cell line (MHCC97L) stably transfected with a control vector (MHCC97L/P), or STC1-overexpressing vector (MHCC97L/S1) with human leukemia monocytic cell line (THP-1) was conducted. We reported that MHCC97L/S1 suppressed the migratory activity of THP-1. Real-time PCR analysis revealed the downregulation of the pro-migratory factors, monocyte-chemoattractant protein receptors, CCR2 and CCR4, and macrophage-migratory cytokine receptor, CSF-1R. Transcriptomic analysis of the THP-1 cells co-cultured with either MHCC97L/P or MHCC97L/S1, detected 1784 differentially expressed genes. The Ingenuity Canonical Pathway analysis predicted that RhoA signaling was associated with the inhibition of the cell migration. Western blot analysis revealed a significant reduction of Ser19-phosphorylation on MLC2, a Rho-A downstream target, in the THP-1 cells. Xenograft tumors derived from MHCC97/S1 in mice showed a remarkable decrease in infiltrating macrophages. Collectively, this is the first report to demonstrate the inhibitory effect of STC1-overexpressing cancer cells on macrophage migration/infiltration. Our data support further investigations on the relationship between tumor STC1 level and macrophage infiltration.

## Introduction

Human stanniocalcin-1 (STC1) is a widely expressed glycoprotein in various tissues and is reported to be involved in many biological functions [1, 2]. Genetic analysis showed that the STC1 gene is located at the metastatic susceptibility locus of 8p, associated with tumor progression and metastases [3, 4]. Considerable numbers of clinical studies reported the dysregulation of STC1 in different types of cancers [2]. Experimental studies using different cancer cell models unraveled the roles of STC1 in cell proliferation/apoptosis, inflammation, migration, and angiogenesis. Notably, STC1 is a downstream target gene of HIF-1α, the key transcriptional factor mediating inflammatory responses and tumor transformation [5–8]. In tumor microenvironment (TME), the dynamics of cancer progression are regulated through the interactions of genetics, immunological, and various micro-environmental factors. Among those, tumor-

Research Grant Council, Hong Kong. https://cerg1.
ugc.edu.hk/. The funders had no role in study
design, data collection and analysis, decision to
publish, or preparation of the manuscript.

**Competing interests:** NO authors have competing
interests.

associated macrophages (TAM) in TME is an essential driver of tumor inflammation and progression. TAM secretes multiple factors depending on the stages of tumorigenesis, exerting a yin-yang influence to govern if the tumor is suppressed or metastasis. The consequence of the effect could also modulate responses of tumor cells to therapeutic treatments.

Hepatocellular carcinoma (HCC) is the seventh most common cancer and the third most common cause of death from cancer worldwide [9]. The cancer is mainly caused by chronic inflammation [10, 11]. TAM infiltration and abundance in HCC was found to be associated with either good (<20%) or poor (>80%) prognosis [12–15]. The inconsistent outcomes may be explained by the anti- or pro-inflammatory actions of TAMs at different stages of carcinogenesis. Our previous study showed that STC1 was upregulated in tumor tissues from the analysis of clinical data of 216 HCC patients [16]. The data indicated that a higher expression of STC1 was associated with poor prognostic outcome. However, a negative correlation (p = 0.008, 216 patient samples) of STC1 expression with tumor size was observed. Using HCC cell-line analysis, we showed that STC1 exerted inhibitory action on the growth of tumor spheroids in culture, reduced the pro-migratory effects of IL-6/IL-8 on HCC, and diminished the development of tumor mass in nude mice model [16]. Our recent study demonstrated that STC1 overexpression in HCC inhibited $p70^{S6K}$/p-rpS6 signaling and energy metabolism to reduce tumor growth [17]. Nevertheless, the relating effects of STC1 and anti-/pro-inflammatory environment on tumor development were not revealed. Since tumor progression is strongly influenced by the infiltrating immune cells, in the present study, the effects of tumor cell-derived STC1 on macrophages' differentiation and function were investigated. Using human leukemia monocytic cell line (THP-1) and STC1-overexpressing MHCC97L, a Boyden chamber co-culture model was used to characterize the effects of HCC-macrophage interactions. Changes in the quantity of infiltrating TAM in STC1-overexpressing HCC derived tumor mass from nude mice were studied.

## Materials and methods

### Cell culture and treatment

Human leukemia monocytic cell line THP-1 (ATCC) was cultured in RPMI 1640 (Gibco, Life Technologies), while human hepatocellular carcinoma cell line MHCC97L (a gift from Dr Nikki Lee, The University of Hong Kong) was cultured in high glucose Dulbecco's modified Eagle's medium (DMEM) (Gibco, Life Technologies). The cells were maintained in a humidified, 5% $CO_2$ incubator at 37˚C. The culture media were supplemented with 10% heat-inactivated fetal bovine serum (Gibco, Life Technologies) and antibiotics (25 U/mL penicillin & 25 μg/mL streptomycin) (Life Technologies). To induce THP-1 differentiation to $M_0$ macrophages, the cells were treated with 5 nM phorbol 12-myristate 13-acetate (PMA) (Abcam).

### Total RNA extraction and real-time PCR

Cellular RNA was extracted by TRIZOL Reagent (Gibco/BRL) according to the manufacturer's instructions. Total RNA with a ratio of $A_{260}/A_{280}$ >1.8 was generally obtained and used to synthesize cDNA using SuperScript VILO Master Mix (Invitrogen; Life Technologies). Real-time PCR was performed using the Fast SYBR Green Master Mix and StepOne Real-Time PCR System (Applied Biosystems). The primer sequences were listed in Table 1.

### Western blot analysis

Cells were lysed in a cold radioimmunoprecipitation assay (RIPA) buffer (150 mM NaCl, 50 mM Tris-HCL, pH 7.4, 2 mM EDTA, 1% NP-40, 0.1% SDS), then sonicated using Bioruptor

**Table 1. List of PCR primers.**

| Gene | Sequence |
| --- | --- |
| GAPDH | 5'-GGACCTGACCTGCCGTCTAG-3', |
| | 5'-TAGCCCAGGATGCCCTTGAG-3' |
| STC1 | 5'-TGAGGCGGAGCAGAATGACT-3', |
| | 5'-CAGGTGGAGTTTTCCAGGCAT-3' |
| TNFα | 5'-GGGCCTGTACCTCATCTACT—3', |
| | 5'-TAGATGGGCTCATACCAGGG-3' |
| IL-6 | 5'-AGCCCACCGGGAACGAAAGA—3', |
| | 5'-TGTGTGGGGCGGCTACATCT-3' |
| CD163 | 5'-CCAACAAGATGCTGGAGTGAC -3', |
| | 5'-TGACAGCACTTCCACATTCAAG -3' |
| CD206 | 5'-GCGGAACCACTACTGACTA—3', |
| | 5'-GTTGTTGGCAGCTTTTCCTC-3' |
| CCR2 | 5'-CCCTGTATCTCCGCCTTCAC-3', |
| | 5'-TGTACTGGGGAAATGCGTCC-3' |
| CCR4 | 5'-AATACAAGCGGCTCAGGTCC-3', |
| | 5'-CTTGCACAGACCTAGCCCAA-3' |
| CSF-1R | 5'-GAACATCCACCTCGAGAAGAAA-3', |
| | 5'-GACAGGCCTCATCTCCACAT-3' |
| EGF | 5'-GTCTTGACTCTACTCCACCCC-3' |
| | 5'-CTCGGTACTGACATCGCTCC-3' |
| CSF-1 | 5'-TGGACGCACAGAACAGTCTC-3' |
| | 5'-ATTCAGTCAAGGGTCTGCGG-3' |
| EGFR | 5'-CCCTGACTCCGTCCAGTATT-3' |
| | 5'-CTGCGTGAGCTTGTTACTCG-3' |
| ITGA1 | 5'-GGGTGCTTATTGGTTCTCCG-3', |
| | 5'-CCTCCATTTGGGTTGGTGAC-3' |
| ITGB8 | 5'-GCATTATGTCGACCAAACTTCA-3', |
| | 5'-GCAACCCAATCAAGAATGTAACT-3' |
| CCL2 | 5'-GATCTCAGTGCAGAGGCTCG-3', |
| | 5'-GGGTCAGCACAGATCTCCTT-3' |
| CCL4 | 5'-CTGCCTCCAGCGCTCTC-3', |
| | 5'-ACCAAAAGTTGCGAGGAAG-3' |
| CCL20 | 5'-GCTGCTTTGATGTCAGTGCT-3', |
| | 5'-GCAGTCAAAGTTGCTTGCTG-3' |
| IL-1α | 5'-TCTTCTGGGAAACTCACGGC-3', |
| | 5'-GTGAGACTCCAGACCTACGC-3' |
| IL-1β | 5'-CTGAGCTCGCCAGTGAAATG-3', |
| | 5'-CATGGCCACAACAACTGACG-3' |
| IL-8 | 5'-AAGCCACCGGAGCACTCCAT -3', |
| | 5'-CACGGCCAGCTTGGAAGTCA-3' |
| CCR5 | 5'-CTATGAGGCAACCACAGGCA-3', |
| | 5'-CCTCTATGGGACCCCTTTGC-3' |
| ITGA3 | 5'-GAGGACATGTGGCTTGGAGT-3', |
| | 5'-GTAGCGGTGGGCACAGAC-3' |
| NGEF | 5'-AGCTTTTCCAGGACAGGACG-3' |
| | 5'-GGAGCTCATAGGAGTTGGGC-3' |

(*Continued*)

**Table 1.** (Continued)

| Gene | Sequence |
|------|----------|
| LPAR5 | 5'-GAGCAACACGGAGCACAG-3', |
|  | 5'-GACAGATGGCTGCCAAGG-3' |
| SEPTIN5 | 5'-CGCCAAAGCTGACTGTCTTG-3' |
|  | 5'-CATCCTCGTCCGAGTCACAC-3' |
| SEPTIN3 | 5'-GTGAGTTTGCCCTGCTTCG-3' |
|  | 5'-GTGGTCGTCATGGGTTACTG-3' |
| SEPTIN11 | 5'-CTTGTTCCAACCACCGCTTG-3' |
|  | 5'-CAGCTGGATGCCTTTCGTTG-3' |
| PLXNA1 | 5'-CTTCGTCATGGACAACGTGC-3', |
|  | 5'-AGCACCGTGTAGTTGAGTCG-3' |
| RND3 | 5'-GTGGGAGACAGTCAGTGTG-3', |
|  | 5'-GAAGTGTCCCACAGGCTC-3' |
| DLC1 | 5'-CACACTGCGTGAAACTGCTC-3' |
|  | 5'-GAAAGAAGTCCGTCCCCGTT-3' |
| ARHGAP5 | 5'-GCGGATTCCATTTGACCTCC-3', |
|  | 5'-TAACTTCCTCCCATGGCTGC-3' |
| CDC42EP2 | 5'-CCGTGGAACGAGTGTTTCCT-3' |
|  | 5'-CGACTGCCACGCTTCAGATA-3' |
| NRP2 | 5'-CGTTCCGGAGAGATTGCCA-3' |
|  | 5'-GCTCCAGTCCACCTCGTAT-3' |

Plus (Diagenode) and centrifuged at 12,000 g for 10 min at 4°C. The cell pellet was discarded. Protein concentration of the supernatant was measured by DC Protein Assay Kit II (BioRad) at absorbance 750nm using a microplate reader (BioTek). The sample lysates were resolved in SDS-PAGE and transferred onto a PVDF membrane (BioRad). The membrane was blocked with 5% non-fat milk in PBST for 1 hr and incubated with a primary antibody, followed by an HRP-conjugated secondary antibody (Table 2). Specific bands were visualized using WEST-SAVE Up (AbFrontier).

**Table 2. List of antibodies used.**

| Antibody | Vendor | Cat. No. | Host |
|----------|--------|----------|------|
| β-actin | Sigma-Aldrich | A2228 | Mouse |
| STC1 | Sigma-Aldrich | HPA023918 | Rabbit |
| MYPT1 | Cell Signaling | 2634T | Rabbit |
| p-MYPT1 (Thr696) | Cell Signaling | 5163T | Rabbit |
| MLC2 | Cell Signaling | 8505T | Rabbit |
| p-MLC2 (Ser19) | Cell Signaling | 3671T | Rabbit |
| LIMK1 | Cell Signaling | 3842S | Rabbit |
| p-LIMK1 (Thr508) | Cell Signaling | 3841T | Rabbit |
| Cofilin | Cell Signaling | 5175T | Rabbit |
| p-Cofilin (Ser3) | Cell Signaling | 3313T | Rabbit |
| F4/80 | Invitrogen | 14-4801-82 | Rat |
| Anti-mouse HRP | Cell Signaling | 7076V | Horse |
| Anti-rabbit HRP | Cell Signaling | 7074V | Goat |
| Anti-rat Alexa Fluor 488 | Thermo Fisher Scientific | A-21208 | Donkey |

## Boyden chamber-based co-culture

Lentiviral overexpression of STC1 in MHCC97L cells (MHCC97L/S1) was prepared, as described in our previous study [17]. The wild-type MHCC97L transfected with an empty vector was labeled as MHCC97L/P. For RNA experiments, $8x10^4$ cells were seeded per well of a 12-well plate. For protein experiments, $1.92x10^5$ cells were seeded per well of a 6-well plate. For cell migration assay, $4x10^4$ cells were seeded per well of a 24-well plate. After overnight incubation, the culture was washed twice and replaced by serum-free DMEM. THP-1 cells were seeded at $1.5x10^6$ cells in a 6-well insert for protein extraction, $9x10^5$ cells in a 12-well insert for RNA extraction, and $3x10^5$ cells for migration assay in serum-free RPMI. The cell culture insert (Falcon) of 0.4 μm pore size membrane was used for the RNA and protein experiments. The cell-insert with 8 μm pore size was used for the migration assay. PMA (5 nM) was used to induce THP-1 differentiation. The ROCK inhibitor Y27632 (10 μM, Sigma) was added to THP-1 cells to validate MHCC97L/S1 cells' anti-migratory effects. In some experiments, 200 ng/mL of MCP-1 (Invitrogen) was added to MHCC97L/P and MHCC97L/S1 cells as a chemoattractant to stimulate THP-1 migration. The Boyden chamber experiments were performed and analyzed in triplicates.

For cell migration assay, after 24–48 hr of incubation, THP-1 cells on the top-side of the inserts were removed by cotton swabs. In contrast, cells migrated to the bottom side of inserts were fixed in ice-cold methanol, followed by staining with 0.5% crystal violet (Farco Chemical Supplies) for 10 min at room temperature. Pictures of migrated cells were captured using light microscopy, and the migrated cells were counted from 4–5 random fields within the inserts using Image J.

For transcriptomic analysis of THP-1 cells, the total RNA's quality was analyzed using the Agilent 2100 Bioanalyzer system. Four replicates per treatment with RNA Integrity Number (RIN) > 8 were used for library construction and sequencing at the Beijing Genomics Institute (Wuhan, China) using BGISEQ-500RS sequencer. Single-end reads of 50 bp read-length were sequenced and then trimmed according to BWA's–q algorithm described previously [18]. Quality-trimmed sequence reads were mapped to human genome reference (GRCh38/hg38). Read-count data were then subjected to differential expression analysis using the edgeR package [19]. Genes with FDR < 0.05 were considered as differentially expressed genes (DEGs). The DEGs were subjected to the Kyoto Encyclopedia of Genes and Genomes (KEGG, www.kegg.jp/kegg/kegg1.html) analysis using the Database for Annotation, Visualization, and Integrated Discovery (DAVID) v6.8 [20, 21] for biological processes analysis. Ingenuity Pathway Analysis (IPA, QIAGEN Redwood City, www.qiagen.com/ingenuity) was implemented to identify altered gene networks and canonical pathways. Canonical pathways with p < 0.05 were considered as statistically significant.

## Cytokine antibody array

THP-1 cells co-cultured with MHCC97L/P or MHCC97L/S1 were collected and lysed according to the instruction of the Human Cytokine Antibody Array kit (Abcam). The sample lysates were then centrifuged at 10,000 rpm for 10 min at 4˚C to remove cell debris. Membranes were incubated with a blocking buffer for 45 min, followed by overnight incubation at 4˚C, in 1 ml of 150 μg cell lysate diluted in the blocking buffer. On the next day, the membranes were washed and incubated with biotin-conjugated anti-cytokine and HRP-conjugated streptavidin solutions. Chemiluminescent signals were captured by X-ray film. Data were analyzed using the Image J Protein Array Analyzer. The experiment of the antibody array was performed in triplicates.

## Immunohistochemical staining of tumor samples from a xenograft animal model

Tumor samples derived from the inoculation of STC1-overexpressing MHCC97L/S1 cells and MHCC-97L/P in the nude mice xenograft model of our previous study [17] were fixed in 4% PFA, dehydrated, and embedded in paraffin. In the study, ten 6-week old male Balb-c nude mice (purchased from the University of Hong Kong) were maintained in a pathogen-free room, under controlled temperature (23 ± 1˚C ambient temperature), a 12-h light/dark cycle, food and water *ad libitum*. The animals were kept and maintained with reference to the "Guidelines and Regulations" of the Department of Health, Hong Kong Special Administrative Region. The protocol of Animal Ethics was approved by the Committee on the "Use of Human and Animal Subjects" of the Hong Kong Baptist University (Permit no. 261812). Animal body weights and tumor sizes (measured by a caliper) were monitored twice a week. The tumor volumes were calculated by a standard equation $[(width^2 \times length)/2]$. On day 32 of the post-inoculation, the mice were sacrificed by cervical dislocation to minimize the suffering and distress. Tumor samples were harvested, and tissue sections of 5 μm were prepared. For immunofluorescent staining, the sections were pretreated with boiling citrate buffer (10 mM sodium citrate, 0.05% Tween 20, pH 6) for antigen-retrieval and blocked with 3% goat serum in PBST (0.05% Tween 20 in PBS) buffer to reduce non-specific binding. The tissue sections were incubated at 4˚C overnight with an antiserum, rat anti-F4/80 (1:50), and a 1 hr incubation with goat anti-rat-Alexa Fluor 488 (1:200) at room temperature. The slides were washed 3 times for 15 min in PBST, after each antiserum application. The slides were counterstained with DAPI (Vectashield). Three pairs of tumors derived from MHCC97/S1 and MHCC97/P were used, in which three sections per tumor were analyzed. The total fluorescent count was quantified from the whole tumor sections.

## Statistical analysis

Statistical analysis was conducted by SigmaPlot 12.0. Data were analyzed using ' 'Student's t-test or one-way analysis of variance (ANOVA) followed by Duncan's multiple range test and were presented as mean ± standard deviation (SD). P-value < 0.05 was considered as statistically significant.

## Results

### Effects of MHCC97L/S1 on migration and chemokine expression of THP-1 cells

The Boyden chamber assay showed that MHCC97L/S1 inhibited the migration of PMA-treated THP-1 cells (Fig 1A). A similar observation was found in the assay using the monocyte chemoattractant protein MCP-1 (200 ng/ml) (Fig 1B). Given the suppressing effects of MHCC97L/S1 on MCP-1 mediated chemotactic migration, gene expression profiles of THP-1 (Fig 2A) and MHCC97L (Fig 2B) at 48-hr of the co-cultures were examined. THP-1 cells co-cultured with MHCC97L/S1 showed increased expression levels of the M1 differentiation markers (TNFα and IL-6), but a downregulation of M2 marker (CD206) (p < 0.05) (Fig 2A, the upper panel). No significant effects on another M2 marker, CD163 was detected. Moreover, the mRNA levels of the C-C chemokine receptors (CCR2 and CCR4) for the migratory cytokines MCP-1 in the THP-1 cells, were significantly reduced (Fig 2A, the lower panel). The transcript levels of the receptor (CSF-1R) for the pro-migratory cytokine CSF1, was also downregulated in the THP-1 cells. In MHCC97L cells, there were no noticeable changes in the

**(A)**

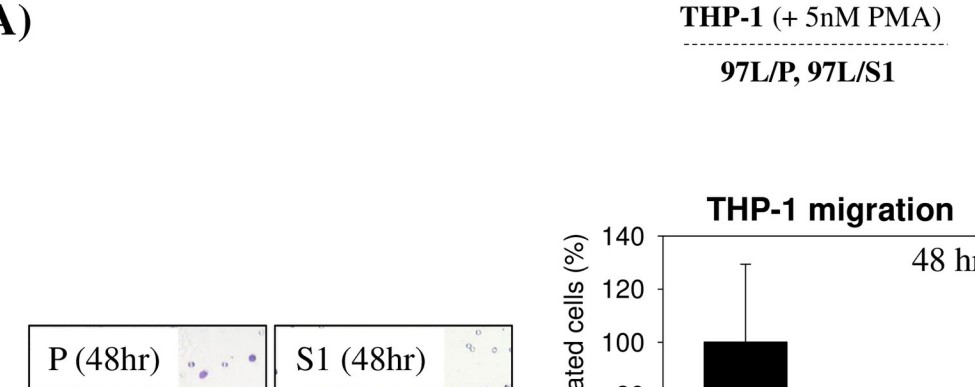

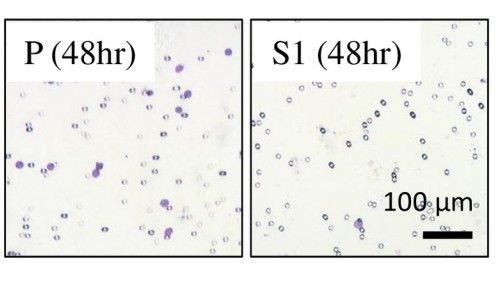

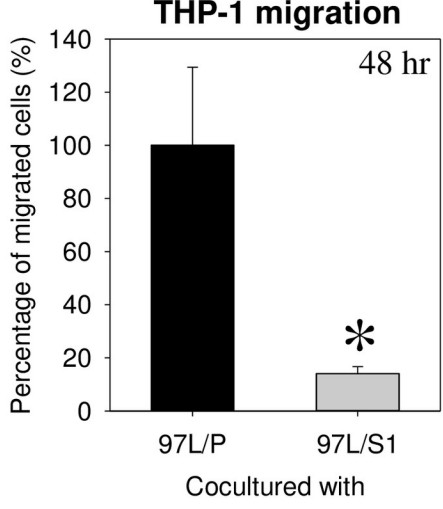

**(B)**

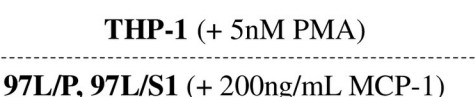

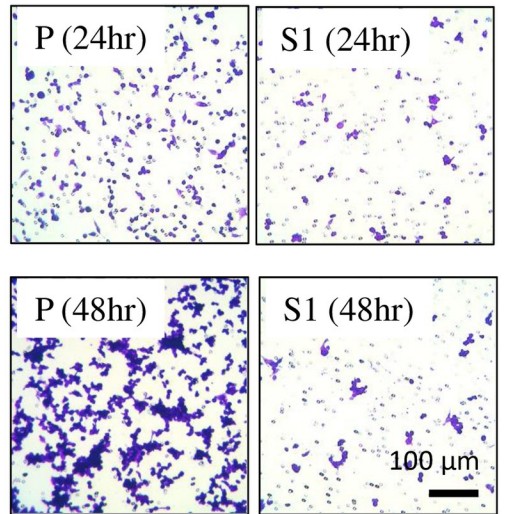

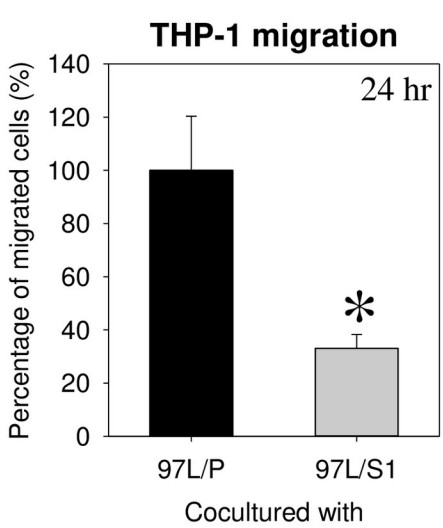

**Fig 1. Effects of STC1-overexpressing MHCC97L (MHCC97L/S1) on THP-1 migration in the Boyden chamber.** PMA (5 nM) treated THP1 cells was seeded in cell culture inserts of 8 μm, and co-cultured with either MHCC97L/S1 or MHCC97L/P for 24 and 48 hr. Migrated THP-1 cells were stained in 0.5% crystal violet and countered using light microscopy. MHCC97L/S1 co-culture inhibited THP-1 migration **(A)** without and **(B)** with 200 ng/mL chemoattractant MCP-1, as compared with MHCC97L/P co-culture. *P < 0.05 as compared with the respective control.

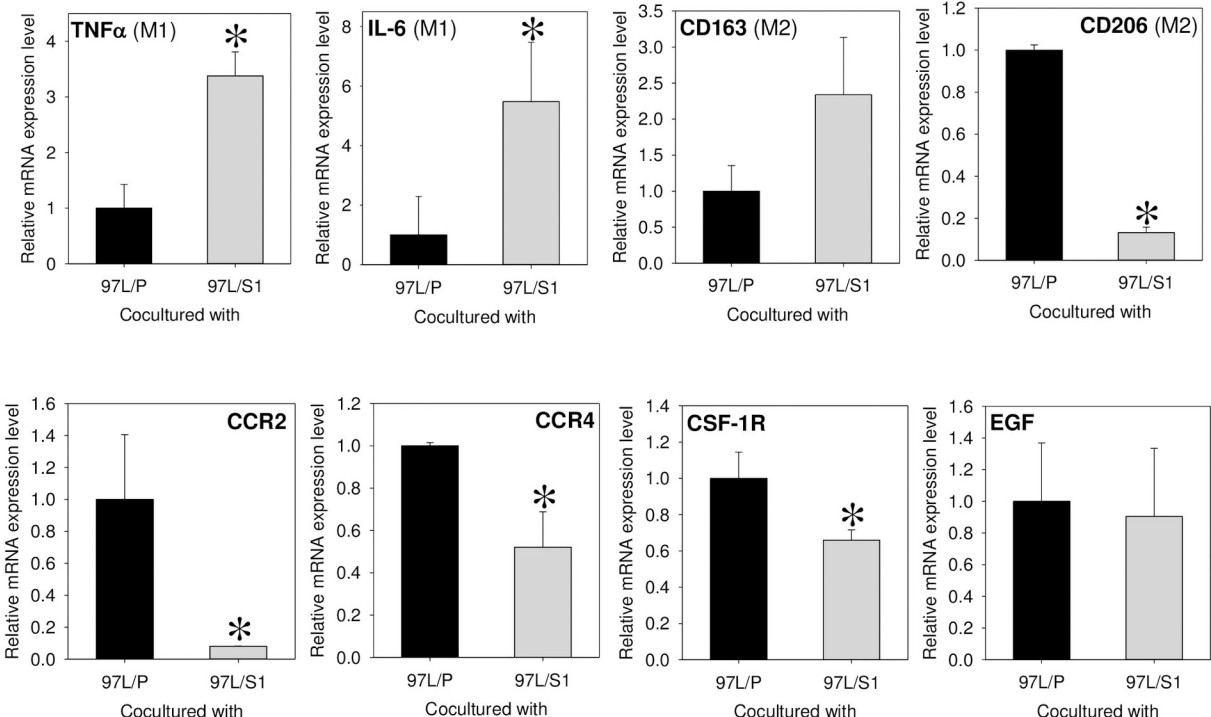

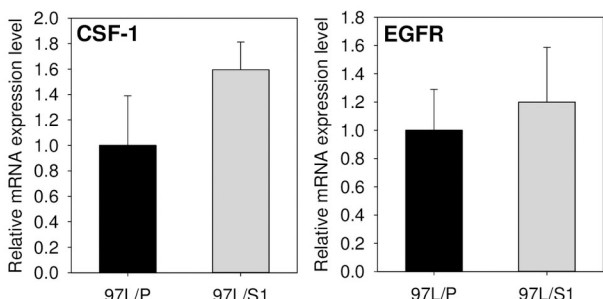

**Fig 2. Gene expressions of THP-1 and MHCC97L in the Boyden chamber.** PMA (5 nM) treated THP1 cells was co-cultured with either MHCC97L/S1 or MHCC97L/P for 48 hr. THP-1 and MHCC97L cells were extracted for real-time PCR analysis. **(A) Gene expressions of THP-1 cells.** The upper panels show that M1 cytokine/marker TNFα and IL-6 were significantly upregulated in MHCC97L/S1 co-culture. The M2 marker CD206 was significantly downregulated. The lower panels show the cytokine/chemokine receptors CCR2, CCR4, and CSF-1R were

significantly downregulated in THP-1 cells. **(B) Gene expressions of MHCC97L.** No significant difference in the gene expression levels of CSF-1 and EGFR were observed in the co-cultures. *P < 0.05 as compared with the respective control.

expression levels of the CSF-1R ligand, CSF-1, and EGFR (Fig 2B). The data supported the observation that STC1-overexpressing-MHCC97L suppressed THP-1 migration.

## Effects of MHCC97L/S1 co-culture on global gene expressions of THP-1 cells

To elucidate the global effects of STC1-overexpressing MHCC97L on THP-1, transcriptomic analysis of the THP-1 cells co-cultured with either MHCC97L/P or MHCC97L/S1 for 24 hr, in the presence of MCP-1 (200 ng/ml) were compared. Quality-trimmed total clean reads of 24.05 M per sample were obtained from THP-1 cells co-cultured with either MHCC97L/P or MHCC97L/S1. A total of 9.6 Gb of clean bases were retrieved, while a total of 17,480 genes were detected, in which 1784 differentially expressed genes (DEGs) (p < 0.05) were identified. There were 820 upregulated and 962 downregulated genes (S1A Fig and S1 Table), subjected to KEGG pathway classification and functional enrichment (S1B Fig). The top three pathways involved signal transduction (363 genes), immune system (268 genes), and cancer overview (201 genes). Pathway functional enrichment results further highlighted the alteration of cytokine-cytokine receptor interaction, TNF signaling pathway, chemokine signaling pathway, and NF-κB signaling pathway (Fig 3A). Based on the altered KEGG pathways, 11 candidate genes associated with cell motility and macrophage functions were chosen (log2 fold change > 1.5, p-value < 0.05, S2 Table) for validation using real-time PCR analysis. Results of PCR validation showed 10 out of 11 candidate genes agreed with the transcriptome data (Fig 3B). To address the effects of STC1-overexpressing MHCC97L on the alteration of cytokine-cytokine receptor interaction, cytokine antibody arrays (n = 3) were used to measure the levels of cytokines in the THP-1 cells. As shown in the representative blot of cytokine antibody array (Fig 4), GRO-α and MCP-2 were significantly upregulated in the THP-1 cells co-cultured with MHCC97L/S1. The levels of IL-1α, IL-8, and RANTES were also increased but statistically not significant. There were no noticeable changes in the levels of TNF-α and IL-6 in the conditioned media of the MHCC97L/P or MHCC97L/S1 cells.

## Effects of MHCC97L/S1 co-culture on RhoA-mediated pathways in THP-1 cells

Ingenuity Canonical Pathway analysis (IPA) revealed that the MHCC97L/S1 co-culture treatment significantly modulated 137 pathways (p < 0.05), in which 106 were stimulated and 31 were inhibited (S3 Table). The canonical pathways were associated with an inhibition of macrophage migration, including (i) RhoA- (z-score: -1.941) and (ii) integrin-signaling pathways (z-score: -0.378). The genes related to the two pathways were listed in Tables 3 and 4. The pathway with the lowest negative z-score, the RhoA signaling pathway, was therefore selected for further investigation. According to IPA analysis, 16 DEGs were associated with the RhoA signaling pathway, whereas most of the dysregulation contributed to a decrease of RhoA activity. Eleven pairs of specific primers for *NGEF*, *LPAR5*, *SEPT5*, *SEPT3*, *SEPT11*, *PLXNA1*, *RND3*, *DLC1*, *ARHGAP5*, *CDC42BP2*, *NRP2* were designed for PCR validation; nine of them matched the results of the transcriptomic analysis (Fig 5A). Western blotting was then conducted to examine the effector proteins of the three major RhoA downstream pathways, including the (i) Rho kinase (ROCK)/MYPT1/MLC2, (i) ROCK/MLC2, and (ii) ROCK/LIMK1/cofilin (Fig 5B). Our data showed a significant increase of MYPT1 phosphorylation at Thr696 and reduced

THP-1 (+ 5nM PMA)

---------------------------------

97L/P, 97L/S1 (+ 200ng/mL MCP-1)

**(A)**

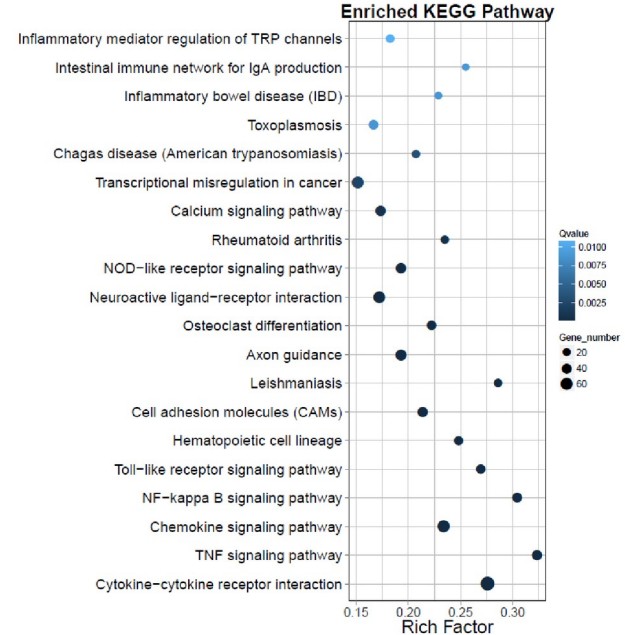

**(B)**

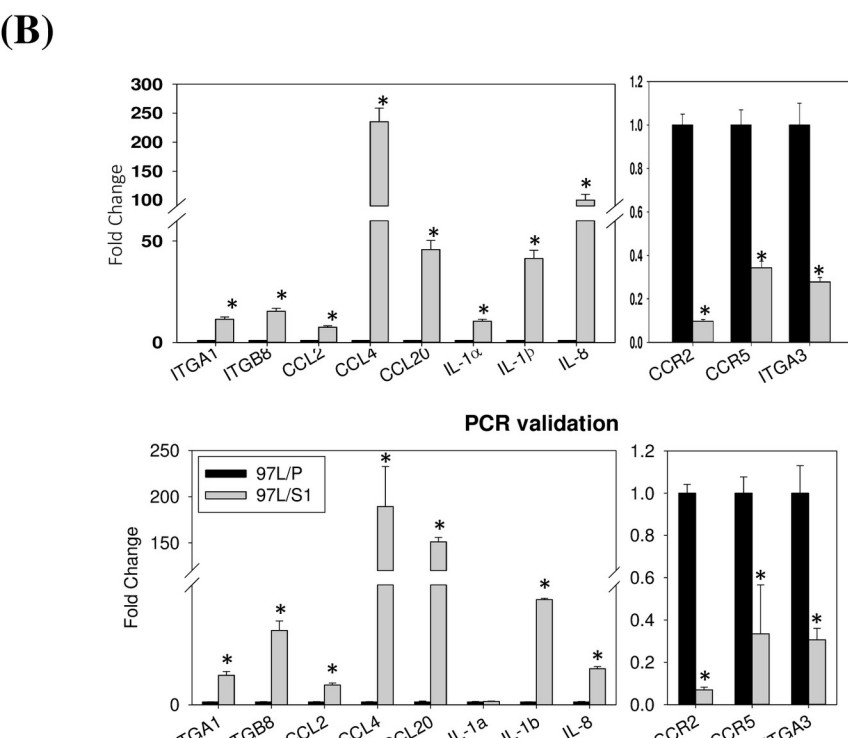

**Fig 3. Transcriptomic analysis of THP-1 in MHCC97L co-culture.** PMA (5 nM) treated THP1 cells was co-cultured with either MHCC97L/S1 or MHCC97L/P for 24 hr. Total RNA of THP-1 cells were extracted for transcriptome. **(A) An enriched KEGG pathway.** X and Y-axes represent the rich factor and pathways, respectively. The rich-factor refers to the

value of the enrichment factor, which is the quotient of the foreground value (number of DEGs) and background value (total gene number). The higher the value, the more significant of the enrichment. The size of the circle indicates the DEG number (a bigger circle refers to a higher number of DEGs). The color of the circle (high: white, low: blue) indicates the q-value. The lower the q-value indicates the higher significance in the enrichment. **(B) A comparison of transcriptome data and qPCR validation** of 11 DEGs (*P < 0.05) as compared with the respective control.

phosphorylated MLC2 at Ser19 in THP-1 cells co-cultured with MHCC97L/S1 (Fig 5C, the left panel). No significant change in the phosphorylation of the effector proteins, LIMK1, and cofi-lin was observed (Fig 5C, the right panel). The western blot data agreed with the Boyden chamber assays and IPA, to support the inhibitory effects of MHCC97L/S1 on the migratory activities of THP-1. Consistently, the use of the ROCK inhibitor Y27632 showed inhibition of THP1 cell migration (S2 Fig).

To illustrate if the inhibitory effects could be verified *in vivo*, tumor mass derived from the inoculation of MHCC97/P and MHCC97/S1 cells in nude mice for 32 days, were collected for immunohistochemical staining of macrophage differentiation marker F4/80. Tumor tissues

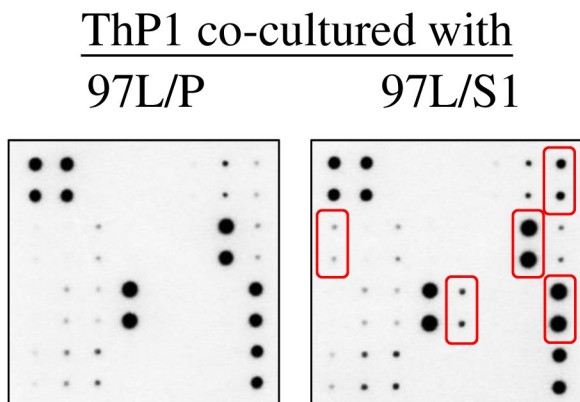

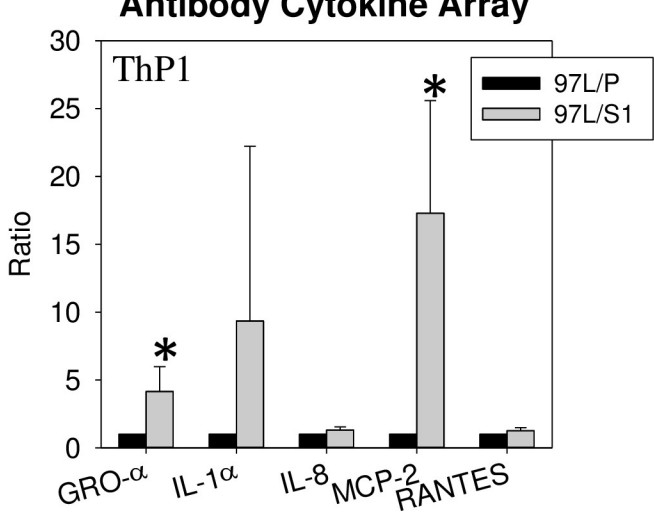

**Fig 4. Human cytokine antibody array.** A comparison of 23 cytokines protein levels using cell lysates of THP-1 co-cultured with MHCC97L/P or MHCC97L/S1. The levels of 5 cytokines, GRO-α, IL-1α, IL-8, MCP-2, and RANTES, were elevated in 97L/S1-cocultured THP-1 cells. The upregulation of GRO-α and MCP-2 were statistically significant (*P < 0.05).

**Table 3. The DEGs of RhoA signaling.** According to IPA analysis on the RhoA pathway, the -log(p-value) is 1.42. The ratio refers to the number of DEGs divided by the total number of genes in the pathway. Z-score shows the likeliness of pathway activation. The value of -1.941 meant the pathway was downregulated.

| -log(p-value) | Ratio | z-score | |
|---|---|---|---|
| 1.42 | 0.129 | -1.941 | |
| **Gene** | **Dysregulation** | **log2 Fold Change** | **P-value** |
| NRP2 | Up | 3.2731 | 7.42E-134 |
| ARHGAP5 | Up | 2.8378 | 7.21E-07 |
| DLC1 | Up | 2.3913 | 2.60E-07 |
| RND3 | Up | 1.5654 | 7.66E-170 |
| PLXNA1 | Up | 1.3669 | 1.38E-203 |
| PLEKHG5 | Down | -3.3501 | 3.93E-98 |
| NGEF | Down | -2.7369 | 4.45E-25 |
| EPHA1 | Down | -2.4429 | 2.44E-12 |
| LPAR5 | Down | -2.4276 | 2.17E-49 |
| CDC42EP2 | Up | 2.9040 | 9.48E-73 |
| CDC42EP3 | Up | 1.0467 | 1.09E-66 |
| SEPT3 | Down | -1.2464 | 3.32E-33 |
| SEPT4 | Down | -2.5227 | 5.93E-13 |
| SEPT5 | Down | -1.9266 | 4.69E-65 |
| SEPT6 | Down | -1.1142 | 6.99E-61 |
| SEPT11 | Up | 1.0553 | 1.46E-111 |

derived from MHCC97/S1 cells (357.5 ± 230.1 mm$^3$) showed significantly less tumor volume as compared with that from MHCC97L/P cells (1236.1 ± 902.9 mm$^3$), as reported in our previous study [17]. Fig 6 shows the representative staining of the tumor sections. The data revealed a higher percentage of F4/80 positive cells in the sections of tumors derived from MHCC97/P than MHCC97/S1 cells.

## Discussion

There is growing evidence on the aberrant expression of STC1 in various types of cancers. The higher expression levels of STC1 in human cancer tissues were documented, although some exceptional cases in breast and ovarian cancers were reported [22]. The higher expression levels of STC1 in tumors are mostly associated with poor prognosis; however, the predictive outcomes are still not conclusive. More extensive sample size studies with additional mechanistic data on tumor microenvironment are yet required to reveal STC1 functions in carcinogenesis. Indeed, tumor mass consists of heterogeneous populations of cancer cells and different types of infiltrating immune cells, including tumor-associated macrophages (TAM), dendritic cells, and T/B cells. Tumor progression is strongly influenced by cancer cells' interactions with the infiltrating host cells, which secrete various factors to determine whether the tumor is suppressed or metastasis. The outcome of the interactions could shape therapeutic responses and resistance of cancer cells in TME, indicating the importance of investigating the role of STC1 in tumor-macrophage interaction.

This study demonstrated that STC1-overexpressing human metastasis hepatocellular carcinoma cells (MHCC97L/S1) suppressed the migratory activity of the human monocytic cells (THP-1) and induced the cells towards M1 differentiation. In the Boyden chamber experiments with or without the monocyte chemoattractant protein-1 (MCP-1/CCL2), THP-1 cells co-cultured with STC1-overexpressing cancer cells showed a significant reduction in migration. MCP-1 is a crucial chemokine that stimulates macrophage infiltration [23–25]. Our data

**Table 4. The DEGs of Integrin signaling.** According to IPA analysis on integrin pathway, the–log(p-value) is 2.89. The ratio refers to number of the DEGs divided by total number of genes in the pathway, Z-score shows the likeliness of pathway activation. The value of -0.378 meant the pathway was downregulated.

| -log(p-value) | Ratio | z-score | |
|---|---|---|---|
| 2.89 | 0.142 | -0.378 | |
| **Gene** | **Dysregulation** | **log2 Fold Change** | **P-value** |
| SRC | Up | 4.8298 | 0 |
| ITGB8 | Up | 3.9452 | 8.05E-101 |
| ITGA1 | Up | 3.5224 | 2.76E-105 |
| ARHGAP5 | Up | 2.8378 | 7.21E-07 |
| TSPAN7 | Up | 1.8132 | 3.90E-30 |
| RND3 | Up | 1.5654 | 7.66E-170 |
| PIK3R5 | Up | 1.5611 | 2.20E-182 |
| RHOQ | Up | 1.4861 | 6.32E-187 |
| BCAR1 | Up | 1.3202 | 7.61E-57 |
| ITGA6 | Up | 1.2539 | 1.34E-82 |
| ITGAV | Up | 1.2231 | 1.41E-147 |
| RHOB | Up | 1.2173 | 6.58E-38 |
| MRAS | Up | 1.1614 | 9.02E-165 |
| RAPGEF1 | Up | 1.0105 | 1.64E-139 |
| CAPN3 | Down | -1.0057 | 6.59E-30 |
| TSPAN5 | Down | -1.0102 | 4.24E-10 |
| ACTN1 | Down | -1.0145 | 6.64E-79 |
| FGFR3 | Down | -1.0328 | 0.00094534 |
| ITGAD | Down | -1.0732 | 0.010799 |
| ITGA2B | Down | -1.0764 | 0.01412158 |
| ITGAM | Down | -1.1294 | 4.18E-106 |
| VCL | Down | -1.1310 | 6.46E-107 |
| ITGA7 | Down | -1.2600 | 4.11E-42 |
| ITGB7 | Down | -1.5424 | 9.97E-161 |
| PAK5 | Down | -1.6062 | 3.13E-19 |
| CAPN11 | Down | -1.6997 | 0.0003719 |
| ITGA11 | Down | -1.7111 | 1.54E-41 |
| ITGA3 | Down | -1.8469 | 3.88E-76 |
| PIK3C2B | Down | -2.0411 | 3.21E-83 |
| TSPAN2 | Down | -2.2459 | 6.15E-19 |
| CAPN6 | Down | -2.6236 | 3.86E-14 |

suggested that the co-culture reduced the response of the THP-1 cells to MCP-1. Notably, our follow-up PCR analysis revealed a significant reduction in the expression levels of the chemokine (MCP-1) receptors, CCR2, and CCR4 in THP-1 cells, to support the observation of the reduced migratory responses. A recent report showed a reduction of CCL2/CCR2 signaling inhibited macrophage infiltration and M2-polarization in hepatocellular carcinoma [26]. Consistently, xenograft tumors in mice treated with CCR2 antagonist revealed less macrophage infiltration. Moreover, another report using CCR2-deficient mice showed that macrophages inclined towards M1 polarization and displayed higher levels of inflammatory cytokine [27]. In this study, we demonstrated that MHCC97L/S1 co-cultured THP-1 inclined M1-polarization, showing an increased expression level of the M1 differentiation markers (TNFα and IL-6), but downregulation of M2 marker (CD206). The data of cytokine array supported this notion, in which a significant upregulation of the pro-inflammatory chemokines, GRO-α

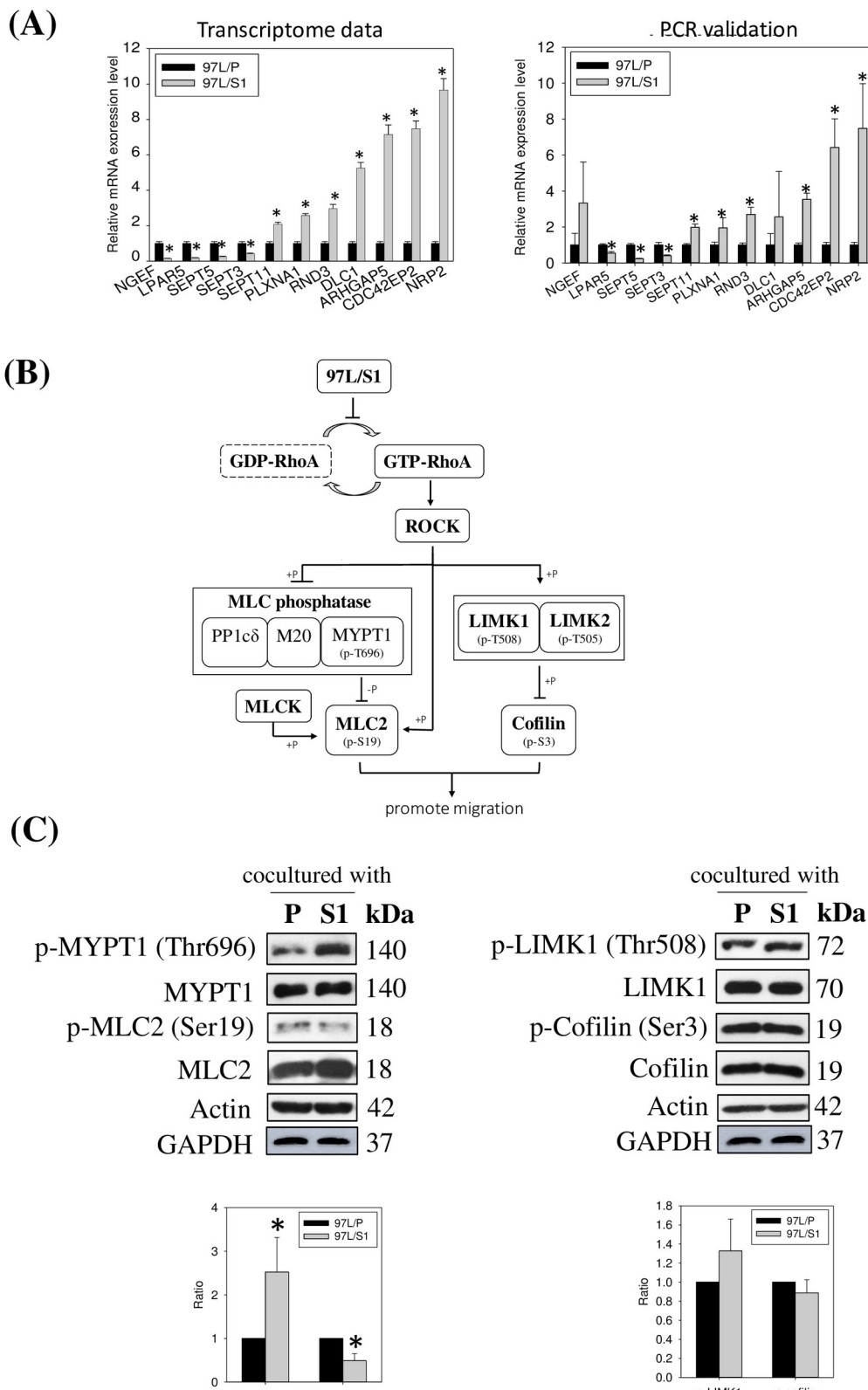

**Fig 5. The validation of the IPA-predicted RhoA pathway suppression in THP-1 cells co-cultured with MHCC97L/S1. (A)** A comparison of 11 RhoA pathway-associated DEGs between transcriptome data and qPCR validation. **(B)** A schematic diagram shows the predicted inhibitory effect of MHCC97L/S1 on the co-cultured THP1

cells. The downstream targets of the RhoA pathway include ROCK/MYPT1/MLC2, ROCK/ MLC2 and ROCK/ LIMK1/cofilin pathways. **(C)** The left panel: In the ROCK/MYPT1/MLC2 pathway, phosphorylation of MYPT (Thr696) was increased, whereas that of MLC2 (Ser19) was decreased (*P < 0.05). The right panel: In the ROCK/ LIMK1/cofilin pathway, there were no noticeable differences in the levels of phosphorylated LIMK1 (Thr508) and cofilin (Ser3) between the two co-cultures.

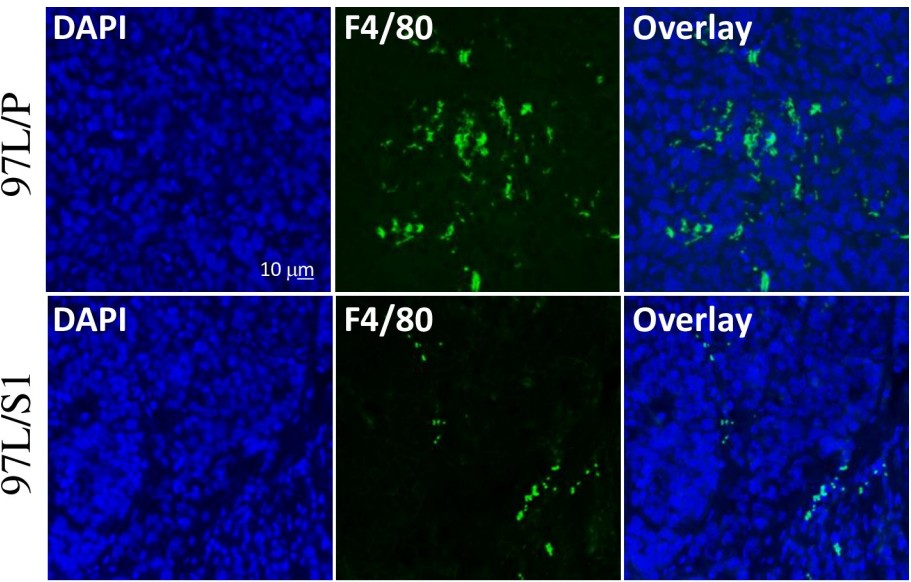

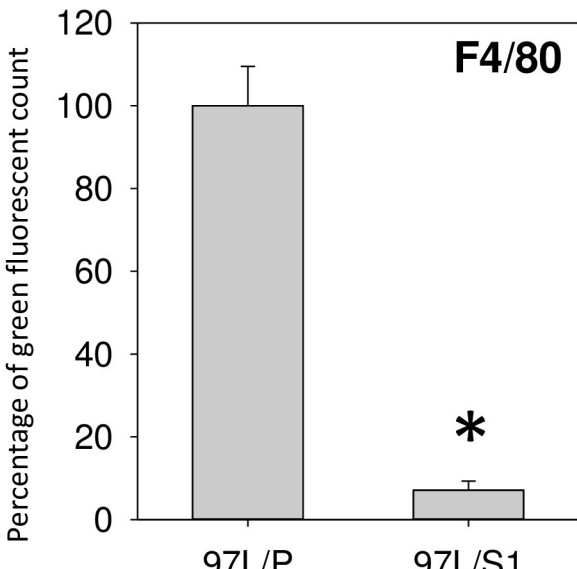

**Fig 6. Immunohistochemical staining of xenograft tumor sections.** Tumor mass derived from the inoculation of MHCC97/P and MHCC97/S1 cells in nude mice were collected for immunohistochemical staining of macrophage-differentiation marker F4/80. Infiltrated macrophages were stained in green (F4/80), and nuclei were in blue (DAPI). The percentage of the green fluorescent count was significantly higher in 97L/P tumors than that in 97L/S1 tumors (*P < 0.05).

(CXCL1) and MCP-2 (CCL8) were detected. In addition to the effect on chemokine pathway, in this study, the expression of a macrophage cytokine receptor, macrophage colony-stimulating factor 1 receptor (CSF-1R), was significantly lower in THP-1 cells co-cultured with MHCC97L/S1 cells. Presumably, the reduction of CSF-1R expression in THP-1 cells led to a decreased response of the cells to CSF-1 secreted by cancer cells [28]. Chakraborty and Kanellis reported a similar observation, in which recombinant STC1 inhibited chemokinesis in macrophage-like RAW264.7 and human monoblast-like (U937) cells towards monocyte chemotactic protein-1 and stromal-derived factor-1α via reduction of intracellular $Ca^{2+}$ levels [29, 30]. Furthermore, STC1 was found to exert inhibitory effects on the transmigration of macrophages and T-lymphocytes, but not on the transmigration of the neutrophils and B-lymphocytes. In the STC1 transgenic mouse model, reduced macrophage infiltration in the glomeruli was observed [31]. Our data collectively demonstrated the effects of STC1 on the reduction of macrophage infiltration and the inclining of THP-1 differentiation to the pro-inflammatory M1 endotype.

Transcriptomic analysis revealed the global effect of STC1-overexpressing MHCC97L/S1 cells on the modulation of cytokine-cytokine receptor interaction, chemokine signaling, and inhibition of the RhoA pathway in THP-1 cells. IPA revealed–that the inhibitory effect was on the RhoA signaling pathway, which was reported to be associated with monocyte/macrophage motility [32–35]. The 16 selected DEGs are known to regulate RhoA activities [36–44]. The upregulation of PLXNA1, NRP2, ARHGAP5, RND3, DLC1, CDC42BP2/CDC42BP3, and the downregulation of EPHA1, NGEF, LPAR5, PLEKHG5, septin 3,4,5,6, identified in this study, were reported to inhibit RhoA activities. Since the relationship between the RhoA pathway and MCP-1-dependent chemotaxis in THP-1 was previously described [45], we decided to characterize this pathway to identify the molecular targets of STC1 in relating to the reduced migratory activity. Apparently, inhibition of RhoA activity would lead to a suppression of its downstream Rho-associated protein kinase (ROCK)-pathways, i) ROCK/MYPT1/ MLC2, ii) ROCK/MLC2, and iii) ROCK/LIMK1/cofilin. The downstream signaling targets, MLC2, and cofilin are responsible for interacting with actin and myosin to promote the formation of stress fibers and to increase cell migration [46, 47]. Western blot analysis revealed the upregulation of phosphorylated MYPT1 (Thr696), the regulatory subunit of MLC phosphatase, to inhibit the catalytic type 1 phosphatase subunit (PP1cσ), resulting in an inhibition of MLC phosphatase activities. Presumably, the reduced MLC phosphatase activity was associated with an increase in the phosphorylation of MLC2. However, our western blot data showed a significant reduction in the level of phosphorylated MLC2. Therefore, a possible regulatory pathway would be via inhibition of RhoA-ROCK-signaling to reduce MLC2 phosphorylation directly. Shi and co-workers (2013) using ROCK2$^{-/-}$ mice-derived mouse embryonic fibroblasts (MEFs) reported that ROCK2$^{-/-}$ reduced phosphorylation of MLC2 [48]. Moreover, the decrease of p-MLC2 level could be due to reducing MLC kinase (MLCK) activities, which is a kinase to phosphorylate MCL2 [49]. Other than ROCK/MYPT1/MLC2 pathway, RhoA regulates the ROCK/LIMK/cofilin pathway. LIMK consists of LIMK1 and LIMK2; those are activated by ROCK via phosphorylation at Thr508 and Thr505, respectively [50, 51]. Previous studies showed their roles in actin cytoskeletal dynamics through phosphorylation of cofilin, one of the actin-binding proteins ADF/cofilin family [52, 53]. For the p-LIMK1/p-cofilin pathway, phosphorylation of cofilin was not noticeably reduced, despite an elevated p-LIMK1 level. Indeed, other than LIMK1, cofilin can also be activated by integrin signaling [54], which was downregulated in this study as predicted by IPA. The observation suggests the possible effect of STC1-overexpressing MHCC97L/S1 cells on the interaction of THP-1 cells with extracellular matrix proteins. It would be interesting to reveal the effects of STC1 on the adhesion of macrophages on different extracellular matrix proteins. With hindsight, the deregulation of

Rho-signaling and the downstream targets would affect cell motility; however, there are multiple pathways involved to determine the functional outcome. Nonetheless, the significant reduction of p-MLC2 agreed with the reduced migratory ability of THP-1 cells co-cultured with MHCC97L/S1.

In addition to the *in vitro* experiments, an animal study was conducted to elucidate the role of STC1 in cancer-macrophage interaction at the tumor environment. Xenograft tumors derived from MHCC97/S1 cells in mice showed a remarkable reduction of infiltrating macrophage, which is one of the significant components in TME, to coordinate various aspects of immunity. Macrophages can exert dual influences on tumorigenesis by suppressing or enhancing tumor development via their pro- or anti-inflammatory responses [55]. However, in most solid tumors, increased infiltration of TAMs is known to associate with poor prognosis in cancers [55, 56]. Our previous studies demonstrated that the inoculation of STC1-overexpressing MHCC97L cells in a nude mice xenograft model exhibited a reduction in tumor mass and volume [16, 17]. Using HCC cell-line analysis (Hep3B and MHCC97L), we also showed the inhibitory actions of STC1 on the growth of tumor spheroids in culture [16]. In this study, our data support our previous observations and the aforementioned in vitro experiments, to demonstrate the first time, a reduction of macrophage infiltration in xenograft tumors. It warrants further investigation to elucidate the involvement of various immune cells in TME.

In summary, the present study using the Boyden chamber model showed the suppressive effects of STC1-overexpressing MHCC97L on the migration of THP-1 cells. The significant reduction in the expression levels of the chemokine/cytokine receptors and the inhibition of the ROCK/MLC2 pathway in THP-1 cells were considered as the potential targets of the suppression. The THP-1 cells co-cultured with MHCC97L/S1 were found to be pro-inflammatory, and the cells were inclined toward M1 endotype, contributing to the reduced sizes of xenograft tumors derived from MHCC97/S1 cells in mice.

## Supporting information

**S1 Fig. (A)** MA plot of DEGs. X-axis represents value A (log2 transformed mean expression level). Y-axis represents value M (log2 transformed fold change). Red dots represent up-regulated DEGs. Blue dots represent down-regulated DEGs. Gray points represent non-DEGs. (B) Pathway classification of DEGs. X axis represents number of DEG. Y axis represents functional classification of KEGG. There are seven branches for KEGG pathways: Cellular Processes, Environmental Information Processing, Genetic Information Processing, Human Disease, Metabolism, Organismal Systems and Drug Development (www.kegg.jp/kegg/kegg1.html). (PPTX)

**S2 Fig. Effects of the ROCK inhibitor (Y27632) on THP-1 migration in the Boyden chamber.** PMA (5 nM) treated THP1 cells was seeded in cell culture inserts of 8 μm, and co-cultured with MHCC97L (with 200 ng/mL chemoattractant MCP-1) for 24 hr. Migrated THP-1 cells were stained in 0.5% crystal violet and countered using light microscopy. THP-1 cells treated with the inhibitor Y27632 (10 μM) and co-cultured with MHCC97L/P, showed a significant reduction in the migration (the left panel) as compared with the cells without the inhibitor treatment (the middle panel). The right panel showed the anti-migratory effects of MHCC97L/S1 on THP-1 cells. *P < 0.05 as compared with the respective control. (PPTX)

**S1 Table.**
(XLSX)

**S2 Table.**
(XLSX)

**S3 Table.**
(XLSX)

**S1 Raw images.**
(PPTX)

## Author Contributions

**Conceptualization:** Chris K. C. Wong.

**Data curation:** Cherry C. T. Leung.

**Formal analysis:** Chris K. C. Wong.

**Funding acquisition:** Chris K. C. Wong.

**Methodology:** Cherry C. T. Leung.

**Project administration:** Cherry C. T. Leung.

**Supervision:** Chris K. C. Wong.

**Writing – original draft:** Cherry C. T. Leung.

**Writing – review & editing:** Chris K. C. Wong.

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
