## [Decision Letter · Decision Letter 0]

7 Sep 2020

PONE-D-20-22832

Effects of Stanniocalcin-1 Overexpressing Hepatocellular Carcinoma Cells on Macrophage Migration

PLOS ONE

Dear Dr. Wong,

Thank you for submitting your manuscript to PLOS ONE. After careful consideration, we feel that it has merit but does not fully meet PLOS ONE’s publication criteria as it currently stands. Therefore, we invite you to submit a revised version of the manuscript that addresses the points raised during the review process.

There are many technical issues with the manuscript as described below by the reviewers. Importantly additional methodological details are needed, improved blotting controls, validation of TNF-α and IL-6 levels in the culture media. Evaluation of the effect of Rho inhibitors in the assay system to independently confirm the conclusion that Rho is responsible would strengthen the work. There is also need to address data presentation in some figures (eg Bar Graphs). 

We look forward to receiving your revised manuscript.

Kind regards,

Joe W. Ramos, Ph.D.

Academic Editor

PLOS ONE

Journal Requirements:

2. At this time, we request that you  please report additional details in your Methods section regarding animal care, as per our editorial guidelines:

(1) Please state the source and number of mice used in the study  

(2) Please provide details of animal welfare (e.g., shelter, food, water, environmental enrichment)

(3) Please describe any steps taken to minimize animal suffering and distress, such as by administering anaesthesia  

(4) Please include the method of euthanasia  

(5) Please describe the post-operative care received by the animals, including the frequency of monitoring and the criteria used to assess animal health and well-being.

Thank you for your attention to these requests.

3. Please provide additional information about each of the cell lines used in this work, including the source and any quality control testing procedures (authentication, characterisation, and mycoplasma testing). For more information, please see http://journals.plos.org/plosone/s/submission-guidelines#loc-cell-lines.

4. In the Methods section, please provide the source, product number and any lot numbers of the primary antibodies used for the Western blot analysis in your study.

5. At this time, we ask that you please provide scale bars on the microscopy images presented in Figure 1 and 6 and refer to the scale bar in the corresponding Figure legend.

Reviewers' comments:

Reviewer's Responses to Questions

**Comments to the Author**

1. Is the manuscript technically sound, and do the data support the conclusions?

Reviewer #1: Partly

Reviewer #2: Yes

2. Has the statistical analysis been performed appropriately and rigorously? 

Reviewer #1: Yes

Reviewer #2: Yes

3. Have the authors made all data underlying the findings in their manuscript fully available?

Reviewer #1: No

Reviewer #2: Yes

4. Is the manuscript presented in an intelligible fashion and written in standard English?

Reviewer #1: Yes

Reviewer #2: Yes

5. Review Comments to the Author

Reviewer #1: Comments of Reviewer

Human stanniocalcin-1 (STC1) is a glycoprotein known to participate in inflammation

and tumor progression. However, its role in cancer-macrophage interaction at the tumor

environment is not known. In this study, the co-culture of the human metastatic

hepatocellular carcinoma cell line (MHCC97L) stably transfected with a control vector

(MHCC97L/P), or STC1-overexpressing vector (MHCC97L/S1) with human leukemia

monocytic cell line (THP-1) was conducted. Authors found that STC1-overexpressing hepatocellular carcinoma (MHCCp7L) suppressed the migratory activity of THP-1, indicating the inhibitory effect of STC1-overexpressing cancer cells on macrophage

migration/infiltration in the relationship between tumor STC1 level and macrophage infiltration.

Major point.

In Boyden chamber-based co-culture, relative TNF-α and IL-6 mRNAs expression were increased in human leukemia monocytic cells (THP-1) by stimulating with PMA, an activator of protein kinase C. Productions of TNF-α and IL-6 in THP-1 may enhanced in THP-1 cells and these factors impact MHCC97L/P or S1 cells. NF-kB signaling may be activated by increased TNF-α and IL-6. Overexpression of STC-1 may suppress activation of NF-kB signaling. This may impact cellular events linked to migration. Authors should be shown these results with additional experiment. Author should determin the levels of TNF-α and IL-6 in condition medium. These experiments may be very important in this study.

Minor point

1. Experimental methods are very poor. This is important to keep reproducibility in experiments by other researchers. Especially, authors should be described cell number used in the experiments of western blot and co-culture.

2. Authors should describe the weight of tumor tissues obtained from mice in immunohistochemical staining.

Reviewer #2: This work reports the role of TME on immune cell infiltration using MHCC97L and THP-1 cell lines. The authors demonstrated that STC-1 overexpressing MHCC97L cells can inhibit transwell migration of THP-1 and identified various changes in cytokines and gene expression in STC-1 overexpressing cells that may be the cause of this inhibitory action. Interestingly, RhoA signaling pathway is implicated as a plausible mechanism of action of STC1. Although definitive proof and explicit mechanism behind STC1 effects is still missing. These results continues to demonstrate the importance of the TME and establishes a baseline by which STC1 overexpressing cancer cells can affect macrophage infiltration.

Comments –

1. Should include scale bar in figures with representative images (Fig 1, Fig 6).

2. Please include information on how many trials/replicates were performed or analyzed for the migration assay (Fig 1, Fig 2, Fig 3). From how many tumor sections was data in Fig 6 generated and is the data quantified from all fields in a tumor section or from select sections only?

3. I agree from your images that it does appear that 97L/S1 co-culture appears to reduce THP-1 transwell migration, however, I wonder how accurate the data is. Your methods describe that cells were counted by light microscopy. From the images, it looks like there are large masses of cells that migrated through, how were those counted? Also, are these quantification per microscopy field or did you image the entire bottom of the transwell and count all cells?

How were the pathways selected in Fig 3A?

4.Fig 3B top panel has no error bars. Was only one replicate analyzed for this bar graph? I interpreted the method section as having 4 replicates run for the transcriptome analysis. Similarly for Fig 5A.

5. In Fig 5C, is actin the representative loading control for the blot? Since RhoA pathway affects actin dynamics, it may be better to use a different loading control to compare your groups (maybe GAPDH).

6. Since RhoA pathway was proposed, it would be nice to do a confirmatory assay perhaps using an inhibitor of RhoA and demonstrating that THP-1 migration is still inhibited.

7. There are some grammatical errors in the writing and formatting inconsistencies that should be addressed before publication.

8. As a general comment, it would be better to show dots when possible on your bar graph (or use alternative means for data display) to indicate what the values were from each of your analyzed experiments used to generate the mean and SD for the bar graphs. This improves the transparency of the data.

6. PLOS authors have the option to publish the peer review history of their article (what does this mean?). If published, this will include your full peer review and any attached files.

Reviewer #1: No

Reviewer #2: No

---

## [Author Response · Author response to Decision Letter 0]

3 Oct 2020

Editorial 

1. At this time, we request that you please report additional details in your Methods section regarding animal care, as per our editorial guidelines:

(a) Please state the source and number of mice used in the study 

Response: Ten 6-week-old male BALB-c nude mice were used in each group. 

(b) Please provide details of animal welfare (e.g., shelter, food, water, environmental enrichment) 

Response: In the study, the mice were maintained in a pathogen-free room, under controlled temperature (23 ± 1ºC ambient temperature), a 12-h light/dark cycle, and food and water ad libitum.

(c) Please describe any steps taken to minimize animal suffering and distress, such as by administering anaesthesia 

Response: The mice were sacrificed by cervical dislocation to minimize the suffering and distress.

(d) Please include the method of euthanasia 

Response: As abovementioned, the animals were sacrificed by cervical dislocation.

(e) Please describe the post-operative care received by the animals, including the frequency of monitoring and the criteria used to assess animal health and well-being. 

Responses: Body weight and tumor volume was measured twice a week to monitor their health and tumor growth.

3. Please provide additional information about each of the cell lines used in this work, including the source and any quality control testing procedures (authentication, characterisation, and mycoplasma testing). For more information, please see http://journals.plos.org/plosone/s/submission-guidelines#loc-cell-lines.

4. In the Methods section, please provide the source, product number and any lot numbers of the primary antibodies used for the Western blot analysis in your study. 

Response: The information is listed in Table 2. 

5. At this time, we ask that you please provide scale bars on the microscopy images presented in Figure 1 and 6 and refer to the scale bar in the corresponding Figure legend.

Response: Added as per your comment.

Reviewer #1: 

Human stanniocalcin-1 (STC1) is a glycoprotein known to participate in inflammation

and tumor progression. However, its role in cancer-macrophage interaction at the tumor

environment is not known. In this study, the co-culture of the human metastatic

hepatocellular carcinoma cell line (MHCC97L) stably transfected with a control vector

(MHCC97L/P), or STC1-overexpressing vector (MHCC97L/S1) with human leukemia

monocytic cell line (THP-1) was conducted. Authors found that STC1-overexpressing hepatocellular carcinoma (MHCCp7L) suppressed the migratory activity of THP-1, indicating the inhibitory effect of STC1-overexpressing cancer cells on macrophage

migration/infiltration in the relationship between tumor STC1 level and macrophage infiltration.

Major point.

In Boyden chamber-based co-culture, relative TNF-α and IL-6 mRNAs expression were increased in human leukemia monocytic cells (THP-1) by stimulating with PMA, an activator of protein kinase C. Productions of TNF-α and IL-6 in THP-1 may enhanced in THP-1 cells and these factors impact MHCC97L/P or S1 cells. NF-kB signaling may be activated by increased TNF-α and IL-6. Overexpression of STC-1 may suppress activation of NF-kB signaling. This may impact cellular events linked to migration. Authors should be shown these results with additional experiment. Author should determine the levels of TNF-α and IL-6 in condition medium. These experiments may be very important in this study.

Response: Agree with your comments. As indicated in the antibody cytokine arrays (Fig 4), we do not see significant changes in the levels of TNF-α and IL-6 in the conditioned media of the MHCC97L/P or S1 cells. 

Minor point.

1. Experimental methods are very poor. This is important to keep reproducibility in experiments by other researchers. Especially, authors should be described cell number used in the experiments of western blot and co-culture.

Response: Sorry for the unclear information. We have added the details accordingly. 

2. Authors should describe the weight of tumor tissues obtained from mice in immunohistochemical staining.

Response: The description has been added in the revised manuscript.

Reviewer #2: 

This work reports the role of TME on immune cell infiltration using MHCC97L and THP-1 cell lines. The authors demonstrated that STC-1 overexpressing MHCC97L cells can inhibit transwell migration of THP-1 and identified various changes in cytokines and gene expression in STC-1 overexpressing cells that may be the cause of this inhibitory action. Interestingly, RhoA signaling pathway is implicated as a plausible mechanism of action of STC1. Although definitive proof and explicit mechanism behind STC1 effects is still missing. These results continues to demonstrate the importance of the TME and establishes a baseline by which STC1 overexpressing cancer cells can affect macrophage infiltration.

Comments 

1. Should include scale bar in figures with representative images (Fig 1, Fig 6).

Response: Scale bars have been added in the revised manuscript.

2. Please include information on how many trials/replicates were performed or analyzed for the migration assay (Fig 1, Fig 2, Fig 3). From how many tumor sections was data in Fig 6 generated and is the data quantified from all fields in a tumor section or from select sections only?

Response: Sorry for the unclear information. Migration assays were performed and analyzed in triplicates. For tumor sections, fluorescent signals of 3 pairs of tumors, 3 sections per tumor, had been analyzed, all showing higher fluorescent count in 97L/P tumors. The representative data from Fig 6 was generated from one of the pair sets. Fluorescent count was quantified from the whole tumor sections. The information has been added in the revised manuscript.

3. I agree from your images that it does appear that 97L/S1 co-culture appears to reduce THP-1 transwell migration, however, I wonder how accurate the data is. Your methods describe that cells were counted by light microscopy. From the images, it looks like there are large masses of cells that migrated through, how were those counted? Also, are these quantification per microscopy field or did you image the entire bottom of the transwell and count all cells?

Response: Data of migrated cells from Fig 1A and 1B (24 hr) were quantified from 4-5 random fields within the inserts. Sometimes, large masses of cells were observed which made it difficult to count. We estimated the number of migrated cells per unit area of a single cell. For figure 1B, the graph was from the 24hr-experiment.

How were the pathways selected in Fig 3A?

Response: 

As mentioned in the Discussion, RhoA pathway was selected according to the transcriptomic data since it was associated with monocyte/macrophage motility. The 2 pathways (MYPT1/MLC2 and LIMK/cofilin) are two common RhoA downstream pathways related to cell motility. 

4.Fig 3B top panel has no error bars. Was only one replicate analyzed for this bar graph? I interpreted the method section as having 4 replicates run for the transcriptome analysis. Similarly for Fig 5A.

Response: We originally took the means of the bioinformatics data to plot the graph and indicated the significance, as the p values are shown in the supplementary Table 1-DEG.xlsx . We did not add the standard deviations in our previous version. As per your comment, we have added the error bars in the revised figures.

5. In Fig 5C, is actin the representative loading control for the blot? Since RhoA pathway affects actin dynamics, it may be better to use a different loading control to compare your groups (maybe GAPDH).

Response: The samples have been used to run for GAPDH and the data are shown in the revised manuscript.

6. Since RhoA pathway was proposed, it would be nice to do a confirmatory assay perhaps using an inhibitor of RhoA and demonstrating that THP-1 migration is still inhibited.

Response: As per your suggestion, we used the ROCK inhibitor (Y27632) in the THP-1 migration assay. A reduction in ThP-1 cell migration is observed. The data are presented in the supplementary Figure 1.

7. There are some grammatical errors in the writing and formatting inconsistencies that should be addressed before publication.

Response: The manuscript has been edited.

---

## [Decision Letter · Decision Letter 1]

23 Oct 2020

Effects of Stanniocalcin-1 Overexpressing Hepatocellular Carcinoma Cells on Macrophage Migration

PONE-D-20-22832R1

Dear Dr. Wong,

We’re pleased to inform you that your manuscript has been judged scientifically suitable for publication and will be formally accepted for publication once it meets all outstanding technical requirements.

Kind regards,

Joe W. Ramos, Ph.D.

Academic Editor

PLOS ONE

Additional Editor Comments (optional):

Reviewers' comments:

Reviewer's Responses to Questions

**Comments to the Author**

1. If the authors have adequately addressed your comments raised in a previous round of review and you feel that this manuscript is now acceptable for publication, you may indicate that here to bypass the “Comments to the Author” section, enter your conflict of interest statement in the “Confidential to Editor” section, and submit your "Accept" recommendation.

Reviewer #1: All comments have been addressed

Reviewer #2: All comments have been addressed

2. Is the manuscript technically sound, and do the data support the conclusions?

Reviewer #1: Yes

Reviewer #2: Yes

3. Has the statistical analysis been performed appropriately and rigorously? 

Reviewer #1: Yes

Reviewer #2: Yes

4. Have the authors made all data underlying the findings in their manuscript fully available?

Reviewer #1: Yes

Reviewer #2: Yes

5. Is the manuscript presented in an intelligible fashion and written in standard English?

Reviewer #1: Yes

Reviewer #2: Yes

6. Review Comments to the Author

Reviewer #1: This reviewer recommends that the revised manuscript was perfectly revised and this reviewer will be suitable for publication in PLOS ONE.

This manuscript does not have concerns about dual publication, research ethics, or publication ethics.

Reviewer #2: (No Response)

7. PLOS authors have the option to publish the peer review history of their article (what does this mean?). If published, this will include your full peer review and any attached files.

Reviewer #1: **Yes: **Masayoshi Yamaguchi

Reviewer #2: No

---

## [Editor Report · Acceptance letter]

27 Oct 2020

PONE-D-20-22832R1 

Effects of Stanniocalcin-1 Overexpressing Hepatocellular Carcinoma Cells on Macrophage Migration 

Dear Dr. Wong:

I'm pleased to inform you that your manuscript has been deemed suitable for publication in PLOS ONE. Congratulations! Your manuscript is now with our production department. 

Kind regards, 

on behalf of

Dr. Joe W. Ramos 

Academic Editor

PLOS ONE